# Personality traits and substance use among college students in Eldoret, Kenya

**Daniel Waiganjo Kinyanjui[1]\*, Ann Mwangi Sum[2]**

**1** Department of Mental Health and Behavioural Sciences, School of Medicine, College of Health Sciences, Moi University, Eldoret, Kenya, **2** Department of Mathematics, Physics & Computing, School of Sciences and Aerospace Studies, Moi University, Eldoret, Kenya

\* waigakinya@yahoo.com

**Data Availability Statement:** All relevant data are within the paper and its Supporting Information files.

**Funding:** The author(s) received no specific funding for this work.

## Abstract

### Background

There is documented evidence of the increase of alcohol and substance use among college students globally. Increased morbidity and associated maladaptive socio-occupational outcomes of the habit with early dependence and mortality have also been reported. Majority of the substance use related studies conducted in low- and middle- income countries mainly look at health- related risk behaviour control mechanisms that focus on the social environment domain, with few or almost none focusing on those embedded within the person (self-control). This study focuses on the relationship between substance use and personality traits (in the self-control domain), among college students in a low- middle- income country.

### Methods

**Design.** A cross- sectional descriptive study that used the self- administered WHO Model Core and the Big Five Inventory Questionnaires to collect information among students in Colleges and Universities in Eldoret town, Kenya.

**Setting.** Four (1- university campus; 3- non- university) tertiary learning institutions were randomly selected for inclusion.

**Subjects.** Four hundred students, 100 from each of the 4 institutions; selected through a stratified multi-stage random sampling, who gave consent to participate in the study. Associations between various variables, personality traits and substance use were tested using bivariate analysis, while the strength/ predictors of association with substance use was ascertained through multiple logistic regression analyses. A finding of $p \leq 0.05$ was considered statistically significant.

### Results

The median age was 21 years (Q1, Q3; 20, 23), approximately half 203 (50.8%) were male, with majority 335 (83.8%) from an urban residence and only 28 (7%) gainfully employed. The lifetime prevalence of substance use was 41.5%, while that of alcohol use was 36%. For both, a higher mean neuroticism score [substance use- (AOR 1.05, 95%CI; 1, 1.10: p = 0.013); alcohol use- (AOR 1.04, 95%CI; 0.99, 1.09: p = 0.032)] showed increased odds of

**Competing interests:** The authors have declared that no competing interests exist.

lifetime use, while a higher mean agreeableness score [substance use- (AOR 0.99, 95%CI; 0.95, 1.02: p = 0.008); alcohol use- (AOR 0.99, 95%CI; 0.95, 1.02: p = 0.032)] showed decreased odds of lifetime use. A higher mean age (AOR 1.08, 95% CI; 0.99, 1,18: p = 0.02) of the students also showed an 8% increase in odds of lifetime alcohol use. The lifetime prevalence of cigarette use was 8.3%. Higher mean neuroticism (AOR 1.06, 95%CI; 0.98, 1.16: p = 0.041) and openness to experience (AOR 1.13, 95%CI; 1.04, 1.25: p = 0.004) scores showed increased odds of lifetime cigarette smoking, whereas being unemployed (AOR 0.23, 95%CI; 0.09, 0.64: p<0.001) had a decreased odd. Other substances reported included cannabis 28 (7%), sedatives 21 (5.2%), amphetamines 20 (*Catha edulis*) (5%), tranquilizers 19 (4.8%), inhalants 18 (4.5%), cocaine 14 (3.5%), with heroin and opium at 10 (2.5%) each. Among the 13 participants who reported injecting drugs, 10 were female and only 3 were male; this finding was statistically significant (p = 0.042).

## Conclusions

The prevalence of substance use among college and university students in Eldoret is high and associated with high neuroticism and low agreeableness personality traits. We provide directions for future research that will examine and contribute to a deeper understanding of personality traits in terms of evidence- based approach to treatment.

## Introduction

Data from research findings reveal relatively high prevalence rates of alcohol and substance use among college and university students in Kenya. For instance, a 2009 study [1] reported a lifetime substance use prevalence rate of 69.8% among college students in Eldoret, Uasin Gishu County, while an earlier study [2] reported even higher rates; 84.2% for alcohol use, 54.7% for tobacco, and 19.7% for cannabis use in a Kenyan private university. Differences between the geographical locations, time- lines and settings, changes in legislation, methodology used, and the accompanying vulnerabilities could plausibly account for the differing prevalence. Various other studies [3–6] have similarly reported high prevalence rates for substances used in secondary schools and tertiary institutions in Kenya. Quarreling, fights, discord in interpersonal relationships, medical problems, damage or loss of property, accidental injuries, robbery or theft, poor performance at school and unplanned unprotected sex are among some of the documented [1] problems associated with substance use in this cohort. This has serious implications when considering the additional risk of early dependence, mental disorders (including suicide), unplanned pregnancies, risk of sexually transmitted infections- including human immunodeficiency virus (HIV), severe morbidity and other associated maladaptive socio-occupational outcomes of the habit, ultimately leading to early mortality. These problems have a bearing on the disability or premature mortality component of the disability-adjusted life years (DALYS) of this cohort. In fact, estimates show that HIV/AIDS, interpersonal violence, depressive disorders, and anxiety disorders account for 1.97%, 0.85%, 0.84%, and 0.45% of DALYS lost respectively; and they are among the top ten causes of DALYS for those aged 20–24 years in Kenya today [7].

Of note is that majority of the substance use studies [1–6] conducted locally mainly look at health- related risk behaviour control mechanisms that focus on the social environment domain, with almost none focusing on those embedded within the person (self- control). The

only local study that assessed the latter was a 2011 dissertation that determined the prevalence of personality disorders among in- patients with substance use disorders at Kenyan rehabilitation centers [8]. Unlike the current study which focuses on personality traits, that earlier study [8] examined the categorical, pervasive, and inflexible personality disorders. Among the very few similar studies from African countries, a Nigerian study that reported low agreeableness trait and high extraversion trait scores as significantly associated with lifetime alcohol use among undergraduate students [9]. Research [10–16] has additionally shown that certain personality profiles are somewhat predisposed to using substances (generally high neuroticism, high openness to experience, low agreeableness, and low conscientiousness) or even linked to a preferential choice of drug. To date, there has been no local systematic study to validate these findings, yet they are often cited as having universal application.

Increasingly, evidence [17–23] has also suggested flexibility and malleability of these functionally disabling personality traits that may predispose to substance use and other related disorders. For instance, 187 (126 identical and 61 fraternal) twins from the Bielefeld Longitudinal Study of Adult Twins (BiLSAT) were followed up and their personality traits assessed longitudinally over a 10-year period using the NEO-PI. Results revealed "substantial individual differences in the change trajectories of both domain and facet scales". These changes were thought to be more complex and influenced by both genetic as well as environmental factors [18]. A longitudinal study investigating the stability of the Big Five personality traits also showed that the mean- level of extraversion, openness to experience, agreeableness, and conscientiousness increased, whereas that of neuroticism decreased over a 9-year period from age 33 to 42 [19]. Another study revealed that personality traits change throughout the lifespan, with the change more pronounced in the young and elderly, and is partly attributable to social demands and experiences [20]. In a meta- analysis of 92 longitudinal studies covering participants with ages ranging from 10 to 101 years; the results found that significant mean-level change in all trait domains occurred, and that 75% of it was in middle age (40–60) and old age (60+) [21]. The individual differences in personality trait change have also been thought to emanate from the unique life experiences encountered [22]. According to Costa and McCrae [23], there may be better opportunities for establishing rapport, understanding the patient and even selecting the optimal therapy with the use of the NEO-PI, which is a personality trait-based instrument.

This study focuses on the relationship between substance use and personality traits (in the self-control domain), among college students in a low- middle- income country. The benefits' emanating from identifying these self- control mechanisms are twofold; on the one hand it would stimulate further research and provide evidence- based approach to prevention and management of substance use disorders; while on the other improve quality of life and overall outcome, including family and socio-occupational functioning, of those living with the habit.

## Methods

### Design

A cross- sectional descriptive study design involving the administration of two instruments was used. The study was carried out between October and November 2014.

### Site

The study was carried out at four tertiary level institutions in Eldoret town, Uasin Gishu County; situated 320 km North West of Nairobi, the capital city of Kenya. The four learning institutions consisted of the Moi University Medical School town campus (along Nandi Road), offers Medicine, Dentistry, Nursing and Public Health degree courses; a private institution (Alphax College) offering degree, diploma and certificate courses (off Iten Road); and two

government- run institutions; a National Polytechnic (along Kisumu-Kapsabet Road) that offers various technological training courses; and a Technical Training Institute (Rift Valley), off Kaptagat Road, that offers courses at diploma and higher National diploma levels. These four institutions, with a total student population of 10,121, were selected from a sampling frame of all tertiary learning institutions in Eldoret Municipality through simple random sampling. We proceeded as follows. A serialized list of all the tertiary learning institutions in Eldoret Municipality was prepared. Each of the identifying number on the list was then copied onto small individual pieces of paper or tickets. All these tickets were then folded to conceal their content, placed in a bowl, and mixed thoroughly; four tickets were thereafter drawn at random out of the bowl. These represented the four institutions mentioned above.

## Participants

A sample size of 379 students was calculated using Fisher's formula with a confidence interval of 95% and the prevalence of personality disorders at 55.7% [8] (based on the prevalence rate of personality disorders among those using substances from the study by Ongeri LG). However, we decided to recruit a total number of 400 students and a sample of 100 students per institution was selected through a process of stratified multi- stage random sampling. This was done proportionate to the number of students in each year in an institution and thereafter simple random sampling was used to select participants within each year using their student registration numbers.

## Outcome measures

The main outcomes were substance use and personality traits. For substance use; 'have you ever used?' was used to asses lifetime use of any of the substances. In this study, lifetime substance use refers to participants who admitted to having ever used at least one of the substances mentioned in the questionnaire. For personality traits; scale scores were created by averaging the 5-point-likert scores of the items as listed in the manual.

## Data collection tools

This study involved the administration of two instruments.

The World Health Organization (WHO) self- administered Model Core Questionnaire [24] was used to collect information on the lifetime use of commonly abused drugs such as alcohol, solvents, marijuana and tobacco, among others. The WHO Model Core Questionnaire was developed through a cross-cultural field assessment at six sites (one in each of the six WHO regions), i.e., Egypt, Greece, India, Mexico, Malaysia and Zimbabwe. Lifetime use in this study refers to having used the substance at any point in the respondent's life. This instrument has been contextualized; previously used locally (in Kenya) to collect data on substance use [1, 25].

The Big Five Inventory (BFI) is the second instrument [26–28] used in this study and consists of 44 items measuring five dimensions of personality, commonly referred to as the OCEAN personality traits: Openness to experience (9 items); Conscientiousness (9 items); Extraversion (8 items); Agreeableness (9 items); Neuroticism (8 items). Scoring while using the BFI uses a 5-point-likert scale from 1 = 'strongly disagree,' to 5 = 'strongly agree'. One needs to create scale scores by averaging several items as instructed in the manual; R is used to denote reverse- scored items. There is also an SPSS syntax that can be used to score the BFI. The alpha reliabilities of the BFI scales typically range from 0.75 to 0.90 and average above 0.80; three-month test-retest reliabilities range from 0.80 to 0.90, with a mean of 0.85. Validity evidence includes substantial convergent and divergent relations with other Big Five

**Table 1. Summary of BFI traits as described by Sattler and Schunck.**

| BFI traits | Description |
|---|---|
| **Openness to experience** | a person's appreciation of new experiences and stimulation due to being imaginative, creative, unconventional, and emotionally as well as aesthetically sensitive |
| **Conscientiousness** | ability to control behavioral and cognitive impulses "that facilitates task-and goal-directed behavior, such as thinking before acting, delaying gratification, following norms and rules, and planning, organizing, and prioritizing tasks" |
| **Extraversion** | energetic approach towards the world and can be understood as a person's tendency to be outgoing, expressive, active, assertive, cheerful, sociable, and in search of stimulation |
| **Agreeableness** | a person's pro-social and communal orientation and includes a person's tendency to be altruistic, trustworthy, cooperative, considerate, empathic, polite, and modest |
| **Neuroticism** | feelings such as anxiety, nervousness, sadness, and depression and thus reflects a tendency to experience negative emotions. |

instruments as well as with peer ratings. Descriptions of the various OCEAN traits in the BFI have been summarized by Sattler and Schunck [29] (Table 1).

## Ethics and consent

The Institutional Research and Ethics Committee at Moi Teaching and Referral Hospital and Moi University conducted ethical review of the study. Permission was also sought and granted from the Heads of the respective institutions (Reference: IREC/2014/202; *Approval Number*: *00012823*). After assurance of absolute confidentiality, questionnaires were given and administered to the selected participants and thereafter checked for completeness before they were collected upon completion. Written informed consent was obtained from all study participants.

## Data storage and analysis

Collected data was entered into a Microsoft Excel database and analyzed using R. Descriptive statistics were used to compute means, medians and standard deviations for numerical variables as well as frequencies for nominal and ordinal variables. Significance of associations between various variables, personality traits and substance use were tested using bivariate analysis and a finding of $p \leq 0.05$ was considered statistically significant. The variables strength/ prediction of association with personality traits and substance use was also ascertained through multiple logistic regression analyses.

## Results

### Sociodemographic distribution

Four hundred students from the four colleges (100 from each college) namely; Moi University College of Health Sciences (MUCHS), Eldoret National Polytechnic (ENP), Alphax College (AC), and Rift Valley Technical Training Institute (RVTTI), participated in the study. The median age was 21 years (Q1, Q3; 20, 23), approximately half 203 (50.8%) were male, with only 28 (7%) gainfully employed. As shown in Table 2, majority 371 (92.8%) were single, from an urban residence 331 (83.8%) and depended on their families 243 (60.8%) for their source of income. However, about a fifth 70 (17.5%) did not have any source of income.

Analysis to determine the distribution of sociodemographic data by the college attended was additionally done. Those participants from the CHS were significantly (p = 0.002) much younger than those from the other colleges who showed a skewed distribution; including slightly older participants. The median age for the participants from CHS (Q1, Q3; 20, 22) and AC (Q1, Q3; 20, 23)

**Table 2. Sociodemographic characteristics of the participants.**

| Variable | | Frequency N = 400 | Percentage (%) |
|---|---|---|---|
| **Sex** | | | |
| | Male | 203 | 50.8 |
| | Female | 197 | 49.2 |
| **Employment status** | | | |
| | None | 372 | 93.0 |
| | Partial/Full-time | 28 | 7.0 |
| **Residence** | | | |
| | Urban | 335 | 83.8 |
| | Rural | 65 | 16.2 |
| **Marital status** | | | |
| | Never married/Single | 371 | 92.8 |
| | Married/Living as a couple | 23 | 5.7 |
| | Divorced/separated | 6 | 1.5 |
| **Source of Income** | | | |
| | Spouse/Family | 243 | 60.8 |
| | No source | 70 | 17.5 |
| | Salary | 31 | 7.8 |
| | Welfare | 29 | 7.2 |
| | Friends | 26 | 6.5 |
| | Illegal income | 1 | 0.2 |

was 21 years, while that for those from RVTTI (Q1, Q3; 20, 23) and ENP (Q1, Q3; 20, 24) was 22 years. There were significantly (p = 0.02) more female (61%) participants than males at AC as compared to the other colleges; CHS (49%); RVTTI (48%); ENP (39%). Those from ENP had significantly (p = 0,001) lower number of years of formal education; 13 years (Q1, Q3; 12, 15) as compared to the others; AC and RVTTI 14 years (Q1, Q3; 13,15); while CHS had 14 years (Q1, Q3; 14,15). A third (33%) of the participants from RVTTI were from a rural residence, whereas 22%, 6%, and only 4% of the participants at ENP, AC, and CHS respectively, were from a rural residence. This relationship was statistically significant (p = 0.001).

## Personality traits

Personality traits among the participants in this study showed statistically significant differences with respect to gender and the college they attended. For instance, female participants had a statistically significantly (p < 0.001) higher mean score 34.68 (s. d. 6.59) in agreeableness trait as compared to their male 32.09 (s. d. 7.35) counterparts. Those from MUCHS attained statistically significantly (p = 0.005) higher mean conscientiousness scores 27.88 (s. d. 3.11) when compared to the mean scores from the other colleges. Participants from RVTTI showed a significantly (p < 0.001) higher mean agreeableness trait score, 36.11 (s. d. 7.31), and a significantly (p < 0.001) lower mean neuroticism trait score, 19.93 (s. d. 5.22), when compared to the mean scores attained by participants from the other colleges. There was no significant association between personality traits and any of the other sociodemographic variables. Table 3 presents factors associated with personality traits.

## Lifetime substance use

Among the participants in this study, 166 (41.5%) reported having used at least one substance in their entire lifetime. Those with low mean scores on neuroticism trait (p = 0,013), high

**Table 3. Factors associated with personality traits.**

| Trait | Sex | | | College | | | | |
|---|---|---|---|---|---|---|---|---|
| | Male | Female | P-value | AC | MUCHS | ENP | RVTTI | P-value |
| | N = 203 | N = 197 | | N = 100 | N = 100 | N = 100 | N = 100 | |
| **Extraversion** | 26.79 | 27.09 | | 26.72 | 26.97 | 26.43 | 27.63 | |
| Mean (SD) | (4.48) | (5.24) | 0.547 | (5.15) | (4.84) | (4.49) | (4.94) | 0.345 |
| **Agreeableness** | 32.09 | 34.68 | | 32.31 | 32.88 | 32.16 | 36.11 | |
| Mean (SD) | (7.35) | (6.59) | <0.001* | (6.76) | (6.80) | (6.84) | (7.31) | <0.001* |
| **Conscientiousness** | 26.36 | 26.34 | | 25.52 | 27.88 | 25.64 | 26.35 | |
| Mean (SD) | (3.57) | (3.61) | 0.945 | (3.37) | (3.11) | (3.79) | (3.58) | <0.001* |
| **Neuroticism** | 21.24 | 21.39 | | 22.52 | 21.37 | 21.44 | 19.93 | |
| Mean (SD) | (4.43) | (5.78) | 0.771 | (5.39) | (4.95) | (4.69) | (5.22) | 0.005* |
| **Openness** | 34.17 | 34.82 | | 34.84 | 34.84 | 33.97 | 34.30 | |
| Mean (SD) | (4.79) | (4.64) | 0.166 | (4.76) | (4.62) | (4.61) | (4.90) | 0.476 |

*Significant at p≤0.05.

AC–Alphax College

MUCHS–College of Health Sciences, Moi University

ENP–Eldoret National Polytechnic

RVTTI–Rift Valley Technical Training Institute

mean scores on agreeableness trait (p = 0.008), and those from the RVTTI college (p = 0.001) had statistically significantly lower lifetime use of any substance. Interestingly, a comparison of the mean personality trait scores between the different colleges also reveals statistically significant trait score differences (p < 0.001); those from RVTTI showed a much higher mean agreeableness and much lower mean neuroticism personality trait scores as compared to the scores from the other colleges.

Among the 400 participants in this study, 144 (36%) reported lifetime alcohol use. There was no statistically significant variation of alcohol use with gender or any of the other sociodemographic variables. The median age of onset of drinking was 16 (7, 18) years, with the youngest age of onset of drinking being 7 years.

A total of 33 (8.3%) out of the 400 participants in this study admitted to have smoked cigarettes or used any other tobacco at least once in their lifetime. The median age of first cigarette was 16 (8, 18) years, with the youngest age of onset being 8 years. There was no statistically significant variation of cigarette use with any of the socio-demographic variables.

Other substances reported included cannabis 28 (7%), sedatives 21 (5.2%), amphetamines (*Catha edulis*) 20 (5%), tranquilizers 19 (4.8%), inhalants 18 (4.5%), heroin and opium at 10 (2.5%) each, and cocaine 14 (3.5%). As shown in Table 4, among the 13 participants who reported injecting drugs, 10 were female and only 3 were male; this finding was statistically significant (p = 0.042). Among these substances, there was no statistically significant variation with any other socio-demographic variable.

## Predictors of substance use

One hundred and sixty-six participants in this study (41.5%) reported using at least one substance in their lifetime. As shown in Table 5 below, being in the MUCHS (AOR 1.14, 95% CI;0.62, 2.09: p = 0.001) and ENP (AOR 1.55, 95%CI; 0.86, 2.8: p = 0.001) had an increased odds of lifetime substance use of 14% and 55% respectively. However, being in RVTTI (AOR 0.54, 95% CI; 0.28, 1.0: p = 0.001) had a decreased odds of lifetime substance use of 46%, while

**Table 4. Patterns of lifetime use of substances by gender.**

| Substance | Male (N = 203) | Female (N = 197) | Total (N = 400) | p value |
|---|---|---|---|---|
| Any substance | 90 (44.3%) | 76 (38.6%) | 166 (41.5%) | 0.243 |
| Alcohol | 79 (38.9%) | 65 (33.0%) | 144 (36.0%) | 0.217 |
| Cigarette and Other | 18 (8.9%) | 15 (7.6%) | 33 (8.2%) | 0.649 |
| Tranquilizers | 11 (5.4%) | 8 (4.1%) | 19 (4.8%) | 0.523 |
| Sedatives | 9 (4.4%) | 12 (6.1%) | 21 (5.2%) | 0.457 |
| Amphetamine | 10 (4.9%) | 10 (5.1%) | 20 (5.0%) | 0.945 |
| Cannabis | 12 (5.9%) | 16 (8.1%) | 28 (7.0%) | 0.386 |
| Hallucinogens | 1 (0.5%) | 6 (3.0%) | 7 (1.8%) | 0.052 |
| Cocaine | 5 (2.5%) | 9 (4.6%) | 14 (3.5%) | 0.252 |
| Heroine | 3 (1.5%) | 7 (3.6%) | 10 (2.5%) | 0.184 |
| Opium | 4 (2.0%) | 6 (3.0%) | 10 (2.5%) | 0.491 |
| Volatile inhalants | 6 (3.0%) | 12 (6.1%) | 18 (4.5%) | 0.130 |
| Injectables | 3 (1.5%) | 10 (5.1%) | 13 (3.2%) | 0.042* |

*Significant at p≤0.05.

being in AC (AOR 1.0) showed no effect. Additionally, those with a much higher mean agreeableness personality trait score (AOR 0.99, 95%CI; 0.95, 1.02: p = 0.008) showed a decreased odds of lifetime substance use of 1%, while those with higher mean neuroticism personality trait score (AOR 1.05, 95%CI; 1, 1.10: p = 0.013) had an increased odds of lifetime substance use of 5%. Apart from the college attended by the participants there was no other statistically significant association between lifetime prevalence of substance use and any of the selected socio-demographic variables.

One hundred and forty- four participants (36%) in this study reported lifetime alcohol use. As shown in Table 5, those from RVTTI (AOR 0.62, 95% CI; 0.32, 1.19: p = 0.005) showed decreased odds of lifetime alcohol use of 38% while again being in AC (AOR 1.0) showed no effect. Being in MUCHS (AOR 1.26, 95% CI; 0.68, 2.36: p = 0.005) as well as being in ENP (AOR 1.68, 95% CI; 0.92, 3.07: p = 0.005) showed an increased odds of lifetime alcohol use of 26% and 68% respectively. Other factors associated with an increase in odds of lifetime alcohol use included a higher mean age (AOR 1.08, 95% CI; 0.99, 1,18: p = 0.02), with an 8% increase; and a higher mean neuroticism personality trait score (AOR 1.04, 95%CI; 0.99, 1.09: p = 0.032), with a 4% increase in odds. A higher mean agreeableness personality trait score (AOR 0.99, 95%CI; 0.95, 1.02: p = 0.032) showed a decreased odds of lifetime alcohol use of 1%.

Some of the predictors of cigarette smoking included being unemployed (AOR 0.23, 95% CI; 0.09, 0.64: p<0.001) which had a decreased odds in lifetime cigarette smoking of 77%, while a higher mean neuroticism personality trait score (AOR 1.06, 95%CI; 0.98, 1.16: p = 0.041) and a higher mean openness to experience personality trait score (AOR 1.13, 95% CI; 1.04, 1.25: p = 0.004) showed increased odd of lifetime cigarette smoking of 6% and 13% respectively. Table 5 above highlights these findings.

## Discussion

This study found a lifetime substance use prevalence rate (41.5%) which is similar to the rate found among high school students (41.5%) during the early nineties in Kenya [3], and much lower than that found (69.8%) by Atwoli et al [1] in the same location five years earlier. A recent systematic review also reported a similar (41.6%) overall prevalence of any substance

**Table 5. Factors associated with substance use.**

| Variable | Any substance use | | | | | Alcohol | | | | | Smoking | | | | |
|---|---|---|---|---|---|---|---|---|---|---|---|---|---|---|---|
| | No (N = 234) | Yes (N = 166) | AOR | 95% CI | P-value | No (N = 256) | Yes (N = 144) | AOR | 95% CI | P-value | No (N = 367) | Yes (N = 33) | AOR | 95% CI | P-value |
| **Gender** | | | | | 0.243 | | | | | 0.217 | | | | | 0.649 |
| Female | 121 (61.4%) | 76 (38.6%) | 1 | | | 132 (67.0%) | 65 (33.0%) | 1 | | | 182 (92.4%) | 15 (7.6%) | | | |
| Male | 113 (55.7%) | 90 (44.3%) | 1.24 | 0.80, 1.91 | | 124 (61.1%) | 79 (38.9%) | 1.23 | 0.79, 1.92 | | 185 (91.1%) | 18 (8.9%) | 0.98 | 0.44, 2.19 | |
| **Campus** | | | | | 0.001* | | | | | 0.005* | | | | | 0.701 |
| AC | 57 (57.0%) | 43 (43.0%) | 1 | | | 65 (65.0%) | 35 (35.0%) | 1 | | | 92 (92.0%) | 8 (8.0%) | 1 | | |
| MUCHS | 56 (56.0%) | 44 (44.0%) | 1.14 | 0.62, 2.09 | | 61 (61.0%) | 39 (39.0%) | 1.26 | 0.68, 2.36 | | 93 (93.0%) | 7 (7.0%) | 1.22 | 0.37, 4.03 | |
| ENP | 47 (47.0%) | 53 (53.0%) | 1.55 | 0.86, 2.80 | | 53 (53.0%) | 47 (47.0%) | 1.68 | 0.92, 3.07 | | 89 (89.0%) | 11 (11.0%) | 1.68 | 0.58, 5.07 | |
| RVTTI | 74 (74.0%) | 26 (26.0%) | 0.54 | 0.28, 1.00 | | 77 (77.0%) | 23 (23.0%) | 0.62 | 0.32, 1.19 | | 93 (93.0%) | 7 (7.0%) | 1.4 | 0.43, 4.59 | |
| **Age** | | | | | 0.08 | | | | | 0.020* | | | | 1.09 | 0.94, 1.27 | 0.365 |
| Median (IQR) | 21 (20,23) | 22 (20,23) | 1.06 | 0.97, 1.16 | | 21 (20,23) | 22 (20,23) | 1.08 | 0.99, 1.18 | | 21 (20,23) | 22 (20,23) | 1.01 | 0.86, 1.21 | |
| **Education** | | | | | 0.178 | | | | | 0.052 | | | | | 0.450 |
| Median (IQR) | 14 (13,15) | 14 (13,15) | 1.02 | 0.93, 1.12 | | 14 (13,15) | 14 (13,15) | 1.03 | 0.94, 1.13 | | 14 (13,15) | 15 (13,15) | 0.23 | 0.09, 0.64 | |
| **Employment** | | | | | 0.179 | | | | | 0.11 | | | | | <0.001* |
| Full/Part time | 13 (46.4%) | 15 (53.6%) | 1 | | | 14 (50.0%) | 14 (50.0%) | 1 | | | 20 (71.4%) | 8 (28.6%) | 1 | | |
| None | 221 (59.4%) | 151 (40.6%) | 0.71 | 0.31, 1.58 | | 242 (65.1%) | 130 (34.9%) | 0.61 | 0.27, 1.38 | | 347 (93.3%) | 25 (6.7%) | 0.23 | 0.09, 0.64 | |
| **Marital** | | | | | 0.706 | | | | | 0.531 | | | | | 0.068 |
| Ever married | 16 (55.2%) | 13 (44.8%) | 1 | | | 17 (58.6%) | 12 (41.4%) | 1 | | | 24 (82.8%) | 5 (17.2%) | | | |
| Never married/single | 218 (58.8%) | 153 (41.2%) | 0.98 | 0.41, 2.43 | | 239 (64.4%) | 132 (35.6%) | 0.96 | 0.40, 2.41 | | 343 (92.5%) | 28 (7.5%) | 0.46 | 0.13, 1.88 | |
| **Extraversion** | | | | | 0.237 | | | | | 0.254 | | | | | 0.802 |
| Median (IQR) | 27 (24,31) | 26 (23,30) | 0.99 | 0.94, 1.04 | | 27 (24,30) | 26 (23,30.3) | 0.99 | 0.94, 1.04 | | 27 (24,30.5) | 27 (23,30) | 0.99 | 0.91, 1.08 | |
| **Agreeableness** | | | | | 0.008* | | | | | 0.032* | | | | | 0.071 |
| Median (IQR) | 35 (29,40) | 33 (28,38) | 0.99 | 0.95, 1.02 | | 34.5 (29,40) | 32.5 (28,38_ | 0.99 | 0.95, 1.02 | | 34 (28,39) | 30 (27,36) | 0.94 | 0.89, 1.01 | |
| **Conscientiousness** | | | | | 0.761 | | | | | 0.682 | | | | | 0.872 |
| Median (IQR) | 27 (24,28) | 26 (24,29) | 1 | 0.94, 1.07 | | 26 (24,28) | 26 (24,29) | 1.02 | 0.96, 1.09 | | 26 (24,29) | 27 (25,28) | 1.02 | 0.91, 1.15 | |
| **Neuroticism** | | | | | 0.013* | | | | | 0.032* | | | | | 0.041* |
| Median | 21 (17,24) | 22 (19,25) | 1.05 | 1.00, 1.10 | | 21.5 (17,24) | 22 (19,25) | 1.04 | 0.99, 1.09 | | 22 (18,24) | 22 (21,26) | 1.06 | 0.98, 1.16 | |
| **Openness** | | | | | 0.283 | | | | | 0.643 | | | | | 0.004* |
| Median (IQR) | 34 (32,38) | 34.5 (32,38) | 1.05 | 1.00, 1.10 | | 34 (32,38) | 34 (32,38) | 1.04 | 0.99, 1.09 | | 34 (32,38) | 38 (34,39) | 1.13 | 1.04, 1.25 | |

*Significant at p≤0.05; AC–Alphax College; MUCHS–College of Health Sciences, Moi University; ENP–Eldoret National Polytechnic; RVTTI–Rift Valley Technical Training Institute

use among adolescents in sub- Saharan Africa [30]. A contemporary Kenyan study [31] among undergraduate students has also reported an overall lifetime substance prevalence of 48.6%; with those from private universities showing a lifetime prevalence rate (41.4%) similar to that in the current study. This decline in the overall use of substances over the five-year period in the same locality mentioned above, may suggest that there could be a positive influence related to that desired outcome. Two issues stand out as plausibly having contributed to the positive change mentioned above. First was the much-needed awareness about the situation regarding substance use among college students in Eldoret, created through the publication by Atwoli et al [1]. Of note too, was the establishment of an Alcohol and Drug Abuse Rehabilitation Unit (ADAR) at the Moi Teaching and Referral Hospital (MTRH) in Eldoret in 2012; three years after the Atwoli et al study [1] and two years prior to the current study. The MTRH is a Multispecialty National Teaching and Referral Hospital in Eldoret that serves the western region of the country, where both studies were conducted. The establishment of the ADAR was additionally preceded by intensive community outreach programs; involving elaborate public education and awareness campaigns on substance use related risks, prevention, and available resources including interventions. However, it was beyond the scope of this study to ascertain some of the explanations given here; hence the differences in methodology, time-lines, settings, and their accompanying vulnerabilities and opportunities cannot be ruled out.

This slow downward shift in prevalence trend is not only unique to the mentioned local studies as it was also observed by the National Authority for the Campaign Against Drug Abuse (NACADA) during their last two (2012 and 2017) Rapid Situation Assessments of Drugs and Substance Abuse in Kenya [32]. The prevalence rate of 41.5% in the current study, despite being lower than the earlier finding [1], may also warrant attention as this was still higher than the general population national rate (37.1%) at the time [32]. The national body (NACADA) also warned that despite the encouraging trend mentioned above, they now encountered more severe forms of dependence [32]. This clearly has major policy implications, including the continued focus on outreach, monitoring and intervention programs especially among this cohort (college attending students) in order to enable feasibility and sustainability of this positive trend. This cohort is unique in that they experience self- navigation into the world without the usual parental guidance, establish new social contacts and perception of life (academic pressure within a socially competitive and demanding environment), coupled with their vulnerability (experimental nature) and a transition that offers greater independence.

The college attended by the participants was the only sociodemographic variable significantly associated with lifetime substance use. Those from the RVTTI college had statistically significantly (p = 0.001) lower lifetime use of any substance, while being in MUCHS or ENP showed increased odds of lifetime substance use. The locations of both MUCHS and ENP can conceivably entertain suggestions that there's easy accessibility and availability of substances in the two colleges. Eldoret National Polytechnic is located in the neighborhood of the largest slum area (Langas) in Eldoret, that has many informal 'outlets' for selling and using substances. The MUCHS is similarly located within a 4-minute walking distance from the Central Business District in Eldoret town and may plausibly be predisposed to such experiences too. Almost half (44.6%) of the college students participating in a recent study [33] in a neighboring county reported that proximity of alcohol selling premises to their college was the reason for drinking alcohol. Slightly more than a third (37.3%) of the students in the same study [33] reported indulging in alcohol as it was readily available within their college. Earlier reports [34] have also shown that a large percentage of alcohol consumed in Kenya comes from small-scale licit or illicit production within the informal sector. Special and unrestricted aggressive advertisements related to proximity may also play a role in providing opportunities for social

learning, affordability, as well as accessibility and availability of substances in both ENP and MUCHS. The RVTTI location has a much lower density, if any, of such 'outlets', and may therefore be protective in terms of social learning opportunities, affordability, accessibility and availability. Being in AC did not have any effect whatsoever on lifetime substance use. There's need to address aspects of promoting drug free environments and enforcing a drug free policy through campaigns around learning institutions with monitoring to ensure compliance.

Although it was beyond the scope of this study to determine the enforcement and implementation of the different institutional alcohol and drug abuse policies, this could most probably vary and significantly affect or contribute to the differences observed in the odds of lifetime substance use among the colleges. Elsewhere, Otieno et al in a 2015 thesis [35] on implementation of alcohol and drug abuse policy in Kenyan universities found that there was lack of resources, training and even screening tools needed to implement policy. The study [35] adopted the Contextual Interaction Theory (CIT) of policy implementation which states that for policy to be successful, resources, training, and clear patterns of communication between actors must be available. Hence resources are clearly needed for implementation; yet in Kenya only 0.01% of the National Health budget is allocated to Mental Health [36]. In fact, Mental Health is severely underfunded in Kenya; the amount to be spent on Mental Health ought to be USD 2.04 per capita per year, but in reality, only USD 0.0012 is spent [36]. This calls for sustained advocacy for funding in Mental Healthcare in Kenya as well as in other underfunded low- and middle- income countries [37].

Students with a much higher neuroticism personality trait score showed increased odds of lifetime substance use while those with a much higher agreeableness personality trait score had decreased odds of lifetime substance use. Interestingly, those from RVTTI showed both protective aspects in their mean trait scores; a much higher agreeableness personality trait mean score and a much lower neuroticism personality trait mean score as compared to those from other colleges; as mentioned earlier students from RVTTI also showed statistically significantly lower lifetime substance use. Generally, research [11–16] has shown that personality traits of those who use substances are commonly characterized as showing high neuroticism, high openness to experience with low agreeableness and low conscientiousness. These concur with the findings in the current study.

Neuroticism is thought to be an internalizing tendency, that manifests as emotional instability coupled with restlessness. These individuals tend to lack the ability and resources to cope with life stressors, thereby increasing their vulnerability and the use of substances as a form of "self- medication"; to forget and escape it all. Agreeableness on the other hand is politeness and cooperativeness. There are vast resources that accompany this trait; including good stable relationships, a trusting and forgiving nature, reliable social support systems, easily available rescue systems that quickly step in to prevent unfavorable negative events or outcomes. Those with a high agreeableness trait will therefore cope very well and have no need for substances while navigating through life. However, the cross-sectional design of the current study does not support causal inference.

In this study female participants showed a statistically significantly higher agreeableness personality trait as compared to their male counterparts. However, contrary to our expectation, there was no statistically significant difference in their pattern of using substances when compared to their male counterparts; as was evidently shown among the colleges. Perhaps experimentation with the need to be accepted or fit into a "sorority" and appear fashionable, coupled with the decay of cultural etiquette among the females in this study could support this finding. In fact, more females as compared to males in this study (16 females vs 12 males) reported using cannabis though this difference was not statistically significant (p = 0.386). As shown in Table 4, a similar trend was also observed regarding the distribution of prescription

and other drug use in this study. This may reflect an increasingly more tolerant attitude towards females, unlike earlier. In the history of most African cultures [38, 39], a young adult woman's role would not be defined in the context of using substances.

Those from MUCHS also showed a statistically significantly higher mean conscientiousness trait score as compared to the others. Facets involved in determining the BFI conscientiousness domain score include; "doing a thorough job, reliable, not careless, organized, not lazy, efficient with perseverance until the task is finished, makes plans and follows through, and one that is not easily distracted". Some authorities [40–44] support malleability or nuances of personality traits and as such would propose the nurturing of these traits in their respective colleges to fit into the required "model" for the intended purpose (training outcome). The alternative would be that the similarity of the traits is inherent [45–47] and as such eventually led them to similar pathways that subsequently converged into similar choices (colleges). However, other factors may be responsible for this finding, and further research is recommended to clarify this. Elsewhere, research [43, 48] has also shown that conscientiousness was more strongly associated with academic and work engagement. A prospective approach and design would perhaps best explain some of the issues raised here.

The prevalence rate of lifetime alcohol use in this study was 36%. This is similar to the rate (36.3%) found among Debre Berhan university students in neighboring Ethiopia [49] and slightly higher to the rate (31.8%) reported by Taiwo et al in their study [9] among undergraduates in the southwestern part of Nigeria. Five years earlier [1], and in the same setting, the lifetime alcohol prevalence rate was 51.9%. Findings from other studies [2, 31, 33, 50] in similar settings have also reported varied (43.2–84.2%) but higher lifetime alcohol use rates. Though lower than what most have found in similar settings, this current rate is still significant as it yet again suggests the possibility of a positive influence as mentioned earlier. Sustainability of this positive influence could hold a strong prospect of success in the prevention and control of alcohol and substance use in the future and needs to be encouraged.

In this study, those from RVTTI had a decreased odds of lifetime alcohol use, while their counterparts from MUCHS and ENP had increased odds of lifetime alcohol use. Those with a higher mean neuroticism personality trait score and a lower mean agreeableness personality trait score also showed increased odds of lifetime alcohol use. A lower mean agreeableness personality trait score was similarly associated with lifetime alcohol use in the Nigerian study mentioned earlier [9]. However, unlike the findings in the current study, higher extraversion personality trait scores were significantly associated with lifetime alcohol use in the Nigerian cohort [9]. Suggesting that cultural influences and preferences in character may also play a role. An earlier meta-analysis by Kotov et al [16] reported low extraversion showing the largest effect size for dysthymic disorder and social phobia. Unfortunately, this study did not include associated psychiatric disorders as an independent variable. The same pernicious influences by these personality traits, that we alluded to earlier, may as well play a significant role in determining this finding.

Additionally, there was an increased odds in lifetime alcohol use among those with a higher mean age, raising the possibility that the rates of alcohol use increase with age. As expected in any society, including among the college student population, drinking alcohol is condoned among those who are much older. Of note too, was that the availability of financial resources appeared not to impact on lifetime alcohol use; as 50% of the gainfully employed had used, whereas 50% had equally never used alcohol. In this study, the median age at first alcoholic drink was 16 years, with the youngest reported age being 7 years; suggesting that the child might have been raised in a family or environment that used traditional African distilling and brewing methods as their means of earning a living. This early exposure is usually associated with an increased risk of dependence and other biological, psychological and social

complications of the habit. Notable though was that there was no statistically significant difference between male and female students regarding their lifetime alcohol use. Like earlier mentioned, this trend suggests a shift and change in expectable cultural roles that may reflect the effects of urbanization and peer influence on alcohol use.

The lifetime prevalence of cigarette smoking in this sample was 8.3%, which is much lower than what was reportedly found in similar settings [1, 2, 31, 33, 50]. In this sample, being unemployed showed decreased odds of lifetime cigarette smoking, while those with a higher mean neuroticism and openness to experience personality trait scores showed increased odds of lifetime cigarette smoking. Unlike the finding in the previous section on those using alcohol, being gainfully employed (finances) increased the odds of cigarette smoking in this study. The facets determining openness to experience domain in the BFI include; "originality, with new ideas, curious, ingenious, a deep thinker, with active imagination, inventive, values artistic and aesthetic experiences, prefers routine, has many artistic ideas, and is sophisticated in art, music, or literature". The median age at first cigarette was again 16 years, with the youngest reported age of smoking at 8 years. This occurs despite the fact that the legal drinking and smoking age is set at 18 years in Kenya [51, 52]. This clearly shows the enormous influence and impact role- modeling has in the lives and development of children in picking up habits and perspectives of the person they look up to. Previous studies [1, 25] have also shown that about a quarter (19.9% to 24.9%) of those using substances were introduced to the habit by members of their nuclear family or close relatives. It is therefore invaluable to involve role-models (family and significant others) and use targeted interventions that will also include these much younger age groups in the campaign against substance use.

Despite the relatively low lifetime prevalence for the use of other substances in the current study, the findings are much higher than those found in an earlier study in the same locality [1]. They are similarly much higher than what others determined in schools, colleges, universities [1, 3, 4, 31, 33, 50] and even among prison inmates in the same region [25]. For instance, the lifetime prevalence reported for sedatives, tranquillizers, cocaine and opioids at the Eldoret G.K prison [25] was 3.8%, 2.3%, 2.3%, and 1.3% respectively as compared to 5.2%, 4.8%, 3.5%, and 5% respectively in the current study. This shows a worrying trend with gravitation towards the use of "hard" (illicit) drugs in this population; a pattern that may further illustrate the effects of urbanization and the implications of a potential overarching "new" culture with respect to the trending substances and potential clashing of cultures in the "global village" concept [53]. It is therefore no surprise that significantly more participants from RVTTI were from a rural residence; they were found to have protective traits and also showed statistically significantly lower substance use. As mentioned earlier [32], the chances of moving from experimentation, abuse, and finally dependence are therefore markedly heightened. An even more disturbing issue is that the findings may suggest increased availability, accessibility, affordability, and acceptability of these illicit substances with subsequent adaptability by this cohort.

Some of the reasons given for substance use in an earlier study [1] among college students in this same region included; to relax (62,2%), to relieve stress (60.8%), desire to experiment (41.9%), peer pressure (38.9%), and to cope with problems (38.9%). These clearly require a customized interventional plan. Contemporary research [33, 54] has also reported that more than 50% of college students access the internet at least once a day and spend approximately 1.6 to 4.5 hours a day online, preferably during the night. This implies that a more attractive focus on intervention, specifically among the younger age groups, would greatly benefit by relying on electronic/ social media applications to complement the other earlier mentioned interventions. This has been tried elsewhere [55] with initial encouraging uptake that points

towards an affordable, self- reliant, healthy living with perceived anonymity; which are appealing preferences when considering interventions for young adults.

## Strengths of the study

This is the first study in the east and central African region to provide epidemiological information on the relationship between personality traits and substance use. Apart from contextualizing the instruments used, the study also generates important information that is useful in implementation of policy, planning, and resource allocation regarding substance use in our country and in the region. The information obtained here is also very useful in informing future research.

## Limitations of the study

Among the limitations encountered in this study was the fact that we used self- administered questionnaires to collect the data that we have presented here. Some of the information, including that regarding substance use, was not validated using any other measures. However, these instruments have been used locally and elsewhere under similar circumstances, and as such the results are comparable to those found in similar studies.

Another key limitation of this study, like mentioned earlier, is the study's cross- sectional design which precludes causal inference and generalizability. Of note too, is the heterogeneity of the institutions that participated in this study pointing to the need of a much larger study to fully describe the substance use pattern among college students in Kenya. However, the information generated from this study is very useful in informing future research.

Information concerning "reasons for using substances", "outcomes of using substances", and associated psychiatric disorder was also not collected in this study. This would have been invaluable in determining if there is an evidence- based relationship between personality traits and psychiatric disorders, reasons for use or complications of use. It would have further enabled comparison with findings from other settings.

## Conclusions and recommendations

This study has demonstrated a high prevalence of substance use among college students in western Kenya. The most commonly used substances were alcohol, cigarettes and cannabis. It has also demonstrated a gravitation towards "hard" (illicit) drugs among college and university students in this low- middle income country; highlighting the risk of severe dependence. Universities and other tertiary-level colleges should increase prevention and targeted interventions that incorporate student support services in order to reduce the associated stigma and improve the opportunities for care. Enculturation of the youth, and especially the girls, in traditional Kenyan values beginning in early childhood and continuing advocacy through mass media; including the internet, would have a sustained positive impact on their life skills. High neuroticism and openness to experience personality trait scores were shown to predisposed to substance use, whereas agreeableness personality trait was found to be protective; hence the need for a prospective study to ascertain pliability of these traits. As such, further research that will examine and contribute to a deeper understanding of personality traits is recommended in the future; and would also be important with regard to the development of evidence- based therapeutic interventions. Sadly, the age of onset of use of substances was reported to be as young as 7-years-old in this study. Health promotion and prevention programs on substance use should therefore begin as early as in lower primary and continue all the way to university, the workplace, and even routinely in community outreach services. This would ultimately lead to a customized interventional plan for those with alcohol and substance use disorders. Follow up

studies on challenges in the implementation of institutional policies on alcohol and substance use would also be invaluable. Finally, there is an urgent need to increase the National Health budget allocation to Mental Health in Kenya to match the international median health financing of USD 2.04 per capita per year.

## Supporting information

**S1 Data.**
(CSV)

## Acknowledgments

The authors would like to acknowledge the heads of the institutions that participated in the study for organizing their students to participate.

## Author Contributions

**Conceptualization:** Daniel Waiganjo Kinyanjui, Ann Mwangi Sum.

**Data curation:** Ann Mwangi Sum.

**Formal analysis:** Daniel Waiganjo Kinyanjui, Ann Mwangi Sum.

**Investigation:** Daniel Waiganjo Kinyanjui, Ann Mwangi Sum.

**Methodology:** Daniel Waiganjo Kinyanjui, Ann Mwangi Sum.

**Project administration:** Daniel Waiganjo Kinyanjui.

**Resources:** Daniel Waiganjo Kinyanjui, Ann Mwangi Sum.

**Supervision:** Daniel Waiganjo Kinyanjui.

**Validation:** Daniel Waiganjo Kinyanjui, Ann Mwangi Sum.

**Writing – original draft:** Daniel Waiganjo Kinyanjui.

**Writing – review & editing:** Daniel Waiganjo Kinyanjui, Ann Mwangi Sum.

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
