## [Decision Letter · Decision Letter 0]

29 Mar 2023

PONE-D-23-03977PERSONALITY TRAITS AND SUBSTANCE USE AMONG COLLEGE STUDENTS IN ELDORET, KENYA.PLOS ONE

Dear Dr. Kinyanjui,

Thank you for submitting your manuscript to PLOS ONE. After careful consideration, we feel that it has merit but does not fully meet PLOS ONE’s publication criteria as it currently stands. Therefore, we invite you to submit a revised version of the manuscript that addresses the points raised during the review process.

We look forward to receiving your revised manuscript.

Kind regards,

Violet Naanyu, PhD

Academic Editor

PLOS ONE

Journal Requirements:

Additional Editor Comments:

Thank you for a study that focusses on a very important matter. Consider the length of your manuscript and the methodological concerns expressed by reviewers. Consider additional biostatistics support/co-author and run (and report) the analyses in a flowing build up from univariate to multivariate analyses. Ensure all your recommendations directly emanate from the study results.

Reviewers' comments:

Reviewer's Responses to Questions

**Comments to the Author**

1. Is the manuscript technically sound, and do the data support the conclusions?

Reviewer #1: Partly

Reviewer #2: Yes

Reviewer #3: Yes

2. Has the statistical analysis been performed appropriately and rigorously? 

Reviewer #1: No

Reviewer #2: Yes

Reviewer #3: Yes

3. Have the authors made all data underlying the findings in their manuscript fully available?

Reviewer #1: Yes

Reviewer #2: Yes

Reviewer #3: Yes

4. Is the manuscript presented in an intelligible fashion and written in standard English?

Reviewer #1: Yes

Reviewer #2: Yes

Reviewer #3: Yes

5. Review Comments to the Author

Reviewer #1: In the background, there is clearly a difference in prevalence of substance use disorders between public colleges and a private university, which are from two separate studies. Give explanations for two differing prevalence.

In the background, substantiate or put into perspective the statement “As shown, these documented problems would clearly have a negative impact on the disability- adjusted life

70 years (DALYs) of this cohort. What of these problems have a bearing on the disability or premature mortality component of the DALYs? Are the DALYs in this cohort known and will they be computed in this study, and if not are there available DALYs estimates from the literature.

In the Background, This sentence needs rephrasing to refer to “of substance use”: “Of note is that majority of the studies [1- 6] conducted locally mainly look at health- related risk behaviour control mechanisms that focus on the social environment domain….”

In background, may need to expound on what different meanings and implications of “

dimensional approach (traits) personality” as opposed to “a categorical (disorders) assessment of personality.”

It is not clear how understanding the relation between personality trait and substance use aids in “improve quality of life and overall outcome, including family and socio-occupational functioning, of those living with the habit.”

In methods, while selection of four learning institutions of heterogeneous levels of learning and students helps accumulate the sample size easily it introduces some variability that may need to be addressed during the statistical analysis.

In methods, stratified multi- stage random sampling, would mean that even the four institutions were being randomly preselected from many other institutions, followed by selection of 100 participants from each one of them. Reading your description, you settled on the four institutions without prior sampling from a list, similar to purposive selection, and so this may simply be described as “Stratified random sampling” because sampling is only at the level of each institution.

In methods, how is the World Health Organization (WHO) self- administered Model Core Questionnaire, different from Alcohol, Smoking and Substance Involvement Screening Test (ASSIST) and has it been tested for validation and reliability in Kenya? Please describe this in the methods if data is available.

I would put the descriptions for the 5 OCEAN traits in the BFI, as by Sattler and Schunck in a table and refer to the table in the methods text, to cut down on word count.

How is the BFI scored? Do you sum up the Likert ratings for each trait, and do you then sum total? Do you have a way of generating discrete categories from these scores? These details are not provided in the methods.

The section referred to as procedures should be something like “Ethics approval” and move the timeline for the study up to where the design was described.

In the sociodemographic table, how do these characteristics differ across the recruiting sites.

The variability in the prevalence of substance use across the recruiting institutions reported in the results may related to the sociodemographic factors above. This is worth exploration and accounting for in the analysis.

The reporting of the results is all mixed or all over the place and needs to follow some sequence, eg what are the study characteristics of the participants; do these characteristics differ according to sites/instructions, what is the prevalence of substance use and of do personality traits, does this prevalence differ by site and by sociodemographic characteristics. Is there an association between substance use and traits of personality disorders; do these associations remain significant after accounting for site heterogeneity and sociodemographic factors.

In the discussion, the lower rates of substance use are attributed to national initiatives to control substance use, but is it possible this could also be due to methodological differences including different settings. It is obviously difficult to establish if the control initiatives worked in the case of your study.

I expect that the discussion should change a bit following the suggested analysis approach and presentation of results.

The discussion is also too long and will need to be shorted a bit.

The author list is too short for a quantitative analysis that involved data collection etc. Have the authors read the ICMJE criteria to be sure that no eligible investigators who meets the authorship criteria was left out.

Reviewer #2: The manuscript is well written, though in the abstract line 40 to 41 and 43 to 44 need to be supported with values of the adjusted odds ratios.

Recommendation on line 52 to 53 is not based on current study findings. The author should focus on recommendations based on the personality traits.

On line 84 till 86 it would be good to explain further on the flexibility and malleability of the personality traits based on previous studies.

Sentence 88 to 90 is unnecessary

In the methods section a general understanding of the different tertiary institutions as regards courses undertaken would shed more light on diversity of the population.

In the discussion section: Line 294 to 295 The Alphax college is not discussed much. Line 318 to 325 seems a repetition. It would be more helpful to discuss on why high agreeableness is protective and vice versa for high neuroticism.

Recommendations in line 458 to 462 do not emanate from the current study findings. You may need to rephrase and mention follow-up studies that look into institutional policies on substance abuse. Finally, which interventional plans are you proposing to be implemented

Reviewer #3: The article presents valuable information, on association of personality traits and substance use. Below are my suggestions on how the manuscript can be improved.

Abstract

The authors can try to summarize to make it shorter, especially the background.

Line 41- ‘…..showed increased odds of lifetime use- not clear lifetime use of which

substance

in the conclusion- change the words ‘drug abuse’ to substance use for uniformity.

Introduction

The last two paragraphs can be combined to one, since they present information that is related.

Also, can leave out the author name for the different citation, since it is already captured in the citation which does not require mention of authors names.

Methods

The subsection on design can come first at the top of the section.

The section labelled design better fits as – outcome measurements/ data collection tools. To first mention and describe the outcome, then describe the data collection tool used.

In the subsection labelled procedure, 6o separate the ethics statement so that it appears as a separate subsection.

Then provide more details on the procedure sufficient enough for the reader to understand and allow replicability of the study.

Results

I suggest to first present the descriptive statistics, then results for the bivariate/ multiple regression analysis. This makes it easier for the reader to follow eg first sees the prevalence/ distribution of use then the associations of substance use with the various factors assessed.

Summarize the information in the narrative to highlight major findings and the refer to the table for more details. Avoid repenting a lot of what is already presented in the tables.

Discussion

Overall the discussion can be made shorter.

In the first paragraph summarize the major findings with regards to the study objectives.

There is the mention of college being more statistically associated with substance use. There is need to explain what this means in terms of specific interventions that can be put in place eg inform policy on treatment/prevention strategies.

Paragraph with lines 296-308: as the authors state, this is not within study objectives, hence should be excluded. If to include, the authors need to incorporate it with specific findings in the study eg data presented in the previous paragraph that talks about difference in locations.

Paragraph with lines 309-317: this paragraph appears not to be linked with any specific study findings. If related to the previous paragraph, it can be combined so that they appear as one paragraph. If not, expand on the discussion on the next paragraph beginning at line 318-326. Also, the authors can expound more on the application of the significance factor with regards to one college -RVTTI- how can this information be applied to improve substance use treatment or prevention strategies among the study population?

In the paragraph beginning line 352- provide more details on the application/relevance of this finding in the study setting.

Paragraph beginning line 418-428: this paragraph gives details of findings from a previous study. the authors need to explain how this is related to the current study.

6. PLOS authors have the option to publish the peer review history of their article (what does this mean?). If published, this will include your full peer review and any attached files.

Reviewer #1: No

Reviewer #2: No

Reviewer #3: No

---

## [Author Response · Author response to Decision Letter 0]

20 Apr 2023

Response to Reviewers

We thank the editor for taking time to review our manuscript; and for making constructive comments and suggestions on how the manuscript can be improved. We have revised the manuscript to address the concerns raised, and below is a point-by-point response to the comments.

A. Editor’s review

Journal Requirements:

-This has been done

-This has been done and highlighted, line 183-184.

-Our study did not include minors

-We are not reporting a retrospective study

-Our study’s minimal underlying data set has been uploaded as supporting information files

-Our study’s minimal underlying data set has been uploaded as supporting information files

Additional Editor Comments:

Thank you for a study that focusses on a very important matter. Consider the length of your manuscript and the methodological concerns expressed by reviewers. Consider additional biostatistics support/co-author and run (and report) the analyses in a flowing build up from univariate to multivariate analyses. Ensure all your recommendations directly emanate from the study results.

- We have put this into consideration

B. Reviewers

1. We thank the reviewer for taking time to review our manuscript; and for making constructive comments and suggestions on how the manuscript can be improved. We have revised the manuscript to address the concerns raised by the reviewer, and below is a highlighted point-by-point response to the comments.

Reviewer #1: In the background, there is clearly a difference in prevalence of substance use disorders between public colleges and a private university, which are from two separate studies. Give explanations for two differing prevalence.

-Differences between the geographical locations, time- lines and settings, possible changes in legislation over time that may influence behaviour, methodology used, and the accompanying vulnerabilities and opportunities could plausibly account for the differing prevalence. This has been highlighted, lines 62-64

In the background, substantiate or put into perspective the statement “As shown, these documented problems would clearly have a negative impact on the disability- adjusted life

70 years (DALYs) of this cohort. What of these problems have a bearing on the disability or premature mortality component of the DALYs? Are the DALYs in this cohort known and will they be computed in this study, and if not are there available DALYs estimates from the literature.

-This has been corrected and highlighted, line 73-78

- We have added the DALYS for this cohort and a reference too, line 75-78; 558- 560

In the Background, This sentence needs rephrasing to refer to “of substance use”: “Of note is that majority of the studies [1- 6] conducted locally mainly look at health- related risk behaviour control mechanisms that focus on the social environment domain….”

-This has been corrected and highlighted, line 79

In background, may need to expound on what different meanings and implications of “

dimensional approach (traits) personality” as opposed to “a categorical (disorders) assessment of personality.”

- This has aptly been done in the ensuing section and highlighted, line 84-85; 88-91

It is not clear how understanding the relation between personality trait and substance use aids in “improve quality of life and overall outcome, including family and socio-occupational functioning, of those living with the habit.”

-This emanates from the successful implementation of the first benefit. It is hoped and entirely conceivable that in the near future there might/ will be innovative and successful ways of approaching and managing these malleable personality traits. The first benefit is therefore to the scientific community, whereas the second is to the patient, their kin, and invariably their ability to function.

In methods, while selection of four learning institutions of heterogeneous levels of learning and students helps accumulate the sample size easily it introduces some variability that may need to be addressed during the statistical analysis.

- It would be important to also mention that it was not possible to compare the students by their various socio- demographic variables prior to selection for inclusion into the study; that information about student populations in the various institutions in Eldoret is not available in the public domain.

-However, the variability has been addressed as suggested and various significant findings by recruiting site/ college have been reported after analysis with the possible implications in terms of interventions, policy, etc., mentioned too. 

In methods, stratified multi- stage random sampling, would mean that even the four institutions were being randomly preselected from many other institutions, followed by selection of 100 participants from each one of them. Reading your description, you settled on the four institutions without prior sampling from a list, similar to purposive selection, and so this may simply be described as “Stratified random sampling” because sampling is only at the level of each institution.

- The four institutions were randomly selected from a list of all 29 similar tertiary level institutions in Eldoret Municipality at the time. This was stated, expounded and highlighted, line 133-138.

In methods, how is the World Health Organization (WHO) self- administered Model Core Questionnaire, different from Alcohol, Smoking and Substance Involvement Screening Test (ASSIST) and has it been tested for validation and reliability in Kenya? Please describe this in the methods if data is available.

-We have been unable to find a way of mentioning ASSIST in the context of the current study’s methodology and manuscript since the WHO self-administered Model Core questionnaire was the instrument that we used in collecting the data. We also are of the opinion that a comparison of the two WHO instruments and aligning them to the current manuscript may also be beyond the scope of this study.

- Regarding the testing for validity and reliability in Kenya; this data was not available. However, the WHO Model Core Questionnaire has been used severally in the same locality. In fact, an earlier study that was carried-out in the same setting and has repeatedly appeared in various sections (literature/ methodology/ discussion) of this manuscript used the WHO Model Core Questionnaire. It maintains and provides a better chance for comparability in substance use research. The published local studies that used the same questionnaire have also been cited and highlighted, lines 161-163

I would put the descriptions for the 5 OCEAN traits in the BFI, as by Sattler and Schunck in a table and refer to the table in the methods text, to cut down on word count.

-This has been corrected and highlighted, line 176

How is the BFI scored? Do you sum up the Likert ratings for each trait, and do you then sum total? Do you have a way of generating discrete categories from these scores? These details are not provided in the methods.

-This has been corrected in the methods and highlighted, lines 167-170

The section referred to as procedures should be something like “Ethics approval” and move the timeline for the study up to where the design was described.

-This has been corrected and highlighted, lines 177; 122

In the sociodemographic table, how do these characteristics differ across the recruiting sites.

-The analysis of the sociodemographic characteristics by the colleges/ recruiting sites has been performed and results included and highlighted, lines 203-214

The variability in the prevalence of substance use across the recruiting institutions reported in the results may related to the sociodemographic factors above. This is worth exploration and accounting for in the analysis.

- This has been done throughout the results section

The reporting of the results is all mixed or all over the place and needs to follow some sequence, eg what are the study characteristics of the participants; do these characteristics differ according to sites/instructions, what is the prevalence of substance use and of do personality traits, does this prevalence differ by site and by sociodemographic characteristics. Is there an association between substance use and traits of personality disorders; do these associations remain significant after accounting for site heterogeneity and sociodemographic factors.

-This has been corrected and presented in the suggested format, lines 193- 289

In the discussion, the lower rates of substance use are attributed to national initiatives to control substance use, but is it possible this could also be due to methodological differences including different settings. It is obviously difficult to establish if the control initiatives worked in the case of your study.

- That and similar other possibilities have been addressed, mentioned and the corrections highlighted, lines 310- 312

I expect that the discussion should change a bit following the suggested analysis approach and presentation of results.

- The discussion largely involves the association of the recruiting sites and other statistically significant factors with the main outcome measures e.g., lines 473-475

The discussion is also too long and will need to be shorted a bit.

- Having completely overhauled the results section as suggested and with the corrections in the discussion section, there appears to be a good flow that covers the study findings. As such, we might not be able to shorten this section further.

The author list is too short for a quantitative analysis that involved data collection etc. Have the authors read the ICMJE criteria to be sure that no eligible investigators who meets the authorship criteria was left out.

-The authors have read the ICMJE criteria and are certain that no eligible investigators that fulfill the authorship criteria have been left out.

2. We thank the reviewer for taking time to review our manuscript; and for making constructive comments and suggestions on how the manuscript can be improved. We have revised the manuscript to address the concerns raised by the reviewer, and below is a highlighted point-by-point response to the comments.

Reviewer #2: The manuscript is well written, though in the abstract line 40 to 41 and 43 to 44 need to be supported with values of the adjusted odds ratios. 

-This has been corrected and highlighted, lines 39- 42

Recommendation on line 52 to 53 is not based on current study findings. The author should focus on recommendations based on the personality traits.

-This has been corrected and highlighted, lines 55- 56

On line 84 till 86 it would be good to explain further on the flexibility and malleability of the personality traits based on previous studies.

-This has been done with changes effected in both the explanation as well as the references; that section has been re-done with explanations based on previous studies, as instructed. This is highlighted in lines 95- 112; 591- 613

Sentence 88 to 90 is unnecessary

-The sentence... “It also incorporates the use of a dimensional approach (traits) as opposed to a categorical (disorders) assessment of personality” …... has been excluded. 

In the methods section a general understanding of the different tertiary institutions as regards courses undertaken would shed more light on diversity of the population.

-This information has been included in the methods section and highlighted, lines 127- 131 

In the discussion section: Line 294 to 295 The Alphax college is not discussed much. Line 318 to 325 seems a repetition. It would be more helpful to discuss on why high agreeableness is protective and vice versa for high neuroticism.

-Being in Alphax college seems to have had no predictive effect on the use of substances and this has been mentioned and highlighted in line 346

- As suggested, line 318 to 325 have been removed and replaced with a discussion on the effects of neuroticism and agreeableness and how they may lead to the use of substances. This has been corrected and highlighted, lines 373-381

Recommendations in line 458 to 462 do not emanate from the current study findings. You may need to rephrase and mention follow-up studies that look into institutional policies on substance abuse. Finally, which interventional plans are you proposing to be implemented. 

-We have rephrased to reflect follow- up studies instead; highlighted, line 533- 535

-This has been corrected; recommendations are now in-line with the study findings; proposed interventional plans have also been included and highlighted, lines 519- 523; 526- 532 

3. We thank the reviewer for taking time to review our manuscript; and for making constructive comments and suggestions on how the manuscript can be improved. We have revised the manuscript to address the concerns raised by the reviewer, and below is a highlighted point-by-point response to the comments.

3. Reviewer #3: The article presents valuable information, on association of personality traits and substance use. Below are my suggestions on how the manuscript can be improved.

Abstract

The authors can try to summarize to make it shorter, especially the background.

- The last sentence in the background has been excluded and the conclusion is shorter too.

Line 41- ‘…..showed increased odds of lifetime use- not clear lifetime use of which

substance

-Line 41 has been corrected and is clearer now with the addition of the AOR; indicating that it is both substance use and alcohol use 

in the conclusion- change the words ‘drug abuse’ to substance use for uniformity.

-The conclusion has been changed and no longer uses the phrase ‘drug abuse’

Introduction

The last two paragraphs can be combined to one, since they present information that is related.

Also, can leave out the author name for the different citation, since it is already captured in the citation which does not require mention of authors names.

- The last two paragraphs in the introduction have been combined to one and highlighted, lines 113- 118

- The author names for the different citations have been left out 

Methods

The subsection on design can come first at the top of the section.

- This has been corrected and the design subsection appears first in the methods section and is highlighted, line 120

The section labelled design better fits as – outcome measurements/ data collection tools. To first mention and describe the outcome, then describe the data collection tool used.

-This has been corrected as instructed and is highlighted, lines 148 and 154

In the subsection labelled procedure, 6o separate the ethics statement so that it appears as a separate subsection.

-This has been corrected and highlighted, line 177

Then provide more details on the procedure sufficient enough for the reader to understand and allow replicability of the study.

-This has been done; with more details especially on the sampling procedure used, that have been included towards the end of the site section and this has been highlighted, lines 134-138

Results

I suggest to first present the descriptive statistics, then results for the bivariate/ multiple regression analysis. This makes it easier for the reader to follow eg first sees the prevalence/ distribution of use then the associations of substance use with the various factors assessed.

Summarize the information in the narrative to highlight major findings and the refer to the table for more details. Avoid repenting a lot of what is already presented in the tables.

-This has been done in the results section, lines 193- 289

Discussion

Overall the discussion can be made shorter.

- Having completely overhauled the results section as suggested and with the corrections in the discussion section, there appears to be a good flow that covers the study findings. As such, we might not be able to shorten this section further.

In the first paragraph summarize the major findings with regards to the study objectives.

- This may entail an additional paragraph at the beginning of the discussion to summarize the major findings. However, there appears to be a good flow of ideas in the manner this section is presented now; following the changes effected in the results section. Additionally, there are requests to shorten the discussion section. 

There is the mention of college being more statistically associated with substance use. There is need to explain what this means in terms of specific interventions that can be put in place eg inform policy on treatment/prevention strategies.

-This has been done and highlighted, lines 344- 348

Paragraph with lines 296-308: as the authors state, this is not within study objectives, hence should be excluded. If to include, the authors need to incorporate it with specific findings in the study eg data presented in the previous paragraph that talks about difference in locations.

-This is a very important paragraph that has a bearing on national as well as local strategies for prevention, including policy implications. The authors had incorporated it with the study findings through the statement… ‘this could most probably vary and significantly affect or contribute to the differences observed in the odds of lifetime substance use among the colleges’ This has been highlighted, lines 350-352

Paragraph with lines 309-317: this paragraph appears not to be linked with any specific study findings. If related to the previous paragraph, it can be combined so that they appear as one paragraph. If not, expand on the discussion on the next paragraph beginning at line 318-326.

-This paragraph is actually complete and compares the current study findings with others elsewhere in the world; and they appear to seemingly concur. This has been highlighted, lines 369-372

Also, the authors can expound more on the application of the significance factor with regards to one college -RVTTI- how can this information be applied to improve substance use treatment or prevention strategies among the study population?

-This has been expounded on and highlighted, lines 344-348; 519-523 

In the paragraph beginning line 352- provide more details on the application/relevance of this finding in the study setting.

-The relevance and applicability of the finding reported in line 352 has been discussed in lines 414- 417

Paragraph beginning line 418-428: this paragraph gives details of findings from a previous study. the authors need to explain how this is related to the current study.

-From the preceding paragraph that talks about a global village concept and borrowing from other studies elsewhere regarding where to easily find and interact with this age group, we are suggesting telemedicine/ teletherapy as a possible and appropriate mode of intervention in this cohort.

---

## [Decision Letter · Decision Letter 1]

10 May 2023

PERSONALITY TRAITS AND SUBSTANCE USE AMONG COLLEGE STUDENTS IN ELDORET, KENYA.

PONE-D-23-03977R1

Dear Dr. Kinyanjui,

We’re pleased to inform you that your manuscript has been judged scientifically suitable for publication and will be formally accepted for publication once it meets all outstanding technical requirements.

Kind regards,

Violet Naanyu, PhD

Academic Editor

PLOS ONE

Reviewers' comments:

All your revisions are well noted and appreciated.

Reviewer's Responses to Questions

**Comments to the Author**

1. If the authors have adequately addressed your comments raised in a previous round of review and you feel that this manuscript is now acceptable for publication, you may indicate that here to bypass the “Comments to the Author” section, enter your conflict of interest statement in the “Confidential to Editor” section, and submit your "Accept" recommendation.

Reviewer #1: All comments have been addressed

Reviewer #2: All comments have been addressed

Reviewer #3: All comments have been addressed

2. Is the manuscript technically sound, and do the data support the conclusions?

Reviewer #1: Partly

Reviewer #2: Yes

Reviewer #3: Yes

3. Has the statistical analysis been performed appropriately and rigorously? 

Reviewer #1: Yes

Reviewer #2: Yes

Reviewer #3: Yes

4. Have the authors made all data underlying the findings in their manuscript fully available?

Reviewer #1: Yes

Reviewer #2: Yes

Reviewer #3: Yes

5. Is the manuscript presented in an intelligible fashion and written in standard English?

Reviewer #1: Yes

Reviewer #2: Yes

Reviewer #3: Yes

6. Review Comments to the Author

Reviewer #1: The explanations and responses given by the authors are satisfactory. I checked that the revisions have also been made in the revised manuscript.

Reviewer #2: (No Response)

Reviewer #3: The authors have addressed all questions previously raised and manuscript formatting meets the journal specifications.

7. PLOS authors have the option to publish the peer review history of their article (what does this mean?). If published, this will include your full peer review and any attached files.

Reviewer #1: No

Reviewer #2: No

Reviewer #3: No

---

## [Editor Report · Acceptance letter]

15 May 2023

PONE-D-23-03977R1 

Personality traits and substance use among college students in Eldoret, Kenya. 

Dear Dr. Kinyanjui:

I'm pleased to inform you that your manuscript has been deemed suitable for publication in PLOS ONE. Congratulations! Your manuscript is now with our production department. 

Kind regards, 

on behalf of

Prof. Violet Naanyu 

Academic Editor

PLOS ONE